# Investigation of Thresholds for Asymmetry Indices to Represent the Visual Assessment of Single Limb Lameness by Expert Veterinarians on Horses Trotting in a Straight Line

**DOI:** 10.3390/ani12243498

**Published:** 2022-12-11

**Authors:** Claire Macaire, Sandrine Hanne-Poujade, Emeline De Azevedo, Jean-Marie Denoix, Virginie Coudry, Sandrine Jacquet, Lélia Bertoni, Amélie Tallaj, Fabrice Audigié, Chloé Hatrisse, Camille Hébert, Pauline Martin, Frédéric Marin, Henry Chateau

**Affiliations:** 1LIM France, Labcom LIM-ENVA, 24300 Nontron, France; 2Ecole Nationale Vétérinaire d’Alfort, USC INRAE-ENVA 957 BPLC, CIRALE, 94700 Maisons-Alfort, France; 3Laboratoire de BioMécanique et BioIngénierie (UMR CNRS 7338), Centre of Excellence for Human and Animal Movement Biomechanics (CoEMoB), Université de Technologie de Compiègne (UTC), Alliance Sorbonne Université, 60200 Compiègne, France

**Keywords:** horse, lameness, symmetry, ROC curves, IMU

## Abstract

**Simple Summary:**

Visual gait evaluation made by the equine veterinarian is an essential part of the diagnosis of locomotor disorders. Measurement of movement asymmetry can provide objective support for diagnosis. However, their interpretation remains complex as horses considered to be healthy may show some degree of asymmetry. This study aims to establish and analyze the threshold values for different indices that can be used to discriminate a healthy horse from a horse considered lame by expert clinicians in their daily practice. At least 88% of healthy horses had an upward range of movement of the withers between −10% and 7% of asymmetry. The withers asymmetry of at least 84% of the forelimb lame horses was out of these thresholds. As well, at least 86% of healthy horses had an upward range of movement of the pelvis between −7% and 18% of asymmetry. At least 83% of the hindlimb lame horses were out of these pelvis asymmetry thresholds. Despite the quite low number of horses included in this study (224), these thresholds provide a first help to avoid overinterpretation of asymmetry when using objective gait analysis systems.

**Abstract:**

Defining whether a gait asymmetry should be considered as lameness is challenging. Gait analysis systems now provide relatively accurate objective data, but their interpretation remains complex. Thresholds for discriminating between horses that are visually assessed as being lame or sound, as well as thresholds for locating the lame limb with precise sensitivity and specificity are essential for accurate interpretation of asymmetry measures. The goal of this study was to establish the thresholds of asymmetry indices having the best sensitivity and specificity to represent the visual single-limb lameness assessment made by expert veterinarians as part of their routine practice. Horses included in this study were evaluated for locomotor disorders at a clinic and equipped with the EQUISYM^®^ system using inertial measurement unit (IMU) sensors. Visual evaluation by expert clinicians allocated horses into five groups: 49 sound, 62 left forelimb lame, 67 right forelimb lame, 23 left hindlimb lame, and 23 right hindlimb lame horses. 1/10 grade lame horses were excluded. Sensors placed on the head (_H), the withers (_W), and the pelvis (_P) provided vertical displacement. Relative difference of minimal (AI-min) and maximal (AI-max) altitudes, and of upward (AI-up) and downward (AI-down) amplitudes between right and left stance phases were calculated. Receiver operating characteristic (ROC) curves discriminating the sound horses from each lame limb group revealed the threshold of asymmetry indice associated with the best sensitivity and specificity. AI-up_W had the best ability to discriminate forelimb lame horses from sound horses with thresholds (left: −7%; right: +10%) whose sensitivity was greater than 84% and specificity greater than 88%. AI-up_P and AI-max_P discriminated hindlimb lame horses from sound horses with thresholds (left: −7%; right: +18% and left: −10%; right: +6%) whose sensitivity was greater than 78%, and specificity greater than 82%. Identified thresholds will enable the interpretation of quantitative data from lameness quantification systems. This study is mainly limited by the number of included horses and deserves further investigation with additional data, and similar studies on circles are warranted.

## 1. Introduction

Movement asymmetry is commonly used as an indicator of locomotor disorders by horses. Indeed, the aim of locomotor examination is to identify any impairment and to locate its source. Currently, lameness is visually evaluated by veterinarians. However agreement between veterinarians about lameness grade assessment is low, particularly for subtle lameness detection [1,2]. Modern gait analysis tools provide quantitative measures of asymmetry. The most versatile tool, the inertial measurement units (IMUs), can be used in a clinical setting [3,4]. The issue about the relationship between visual lameness assessment and gait asymmetries measurement systems has been raised. Asymmetry of vertical displacement of the head and pelvis has shown relevant increase with induced lameness [5,6,7]. But even horses perceived by the veterinarian to be sound have demonstrated physiological asymmetrical gait [8,9,10]. Despite the known capacity of withers asymmetry for detecting compensatory movements, it has been studied relatively less than the head [11,12].

In this context, thresholds of asymmetry parameters which correspond to visual evaluation of lameness by veterinarians have been studied. Asymmetry thresholds of the head (>6 mm) and the pelvic (>3 mm) vertical displacement were used for the first time by McCracken et al. [13]. They were probably based on a confidence interval calculated with two repeated measures on 236 horses [14]. These thresholds have been adjusted to the method of data construction used in other IMU systems [15]. A growing number of studies have used these thresholds as an objective lameness detection [11,12,16]. However numerous asymmetry values of sound horses have been over these thresholds [8,17,18]. This might be explained by undetected subclinical, pain-mediated disease or by biological variation, but no consensus has yet been reached [19,20]. Recently, a discrimination method of statistical analysis was applied on 25 Thoroughbred racehorses to redefine higher thresholds, (14.5 mm for the head and 7.5 mm for the pelvis) [21]. In this study, the focus was on specificity because the objective was to screen horses before racing. These results have given guidelines but require further investigations with heterogeneous horses and lameness types using a clinical environment faced by practitioners.

The goals of this clinical observational study were (i) to establish which asymmetry indices have the best sensitivity and specificity to reflect the visual assessment of single-limb lameness made by expert clinicians as part of their routine practice. (ii) Then, for the relevant indices, the aim was to determine the threshold of lameness detection and lame limb identification. This first study was limited to the following conditions: at trot, in hand, on a straight line and on a hard surface.

## 2. Materials and Methods

This clinical observational retrospective study was approved by the clinical research ethics committee (ComERC n◦2022-01-19).

### 2.1. Horses

This study was conducted on horses (*n* = 224), presented at a clinic for locomotor evaluation from August 2019 until October 2021. The sample was composed of 46% females, 47% geldings and 7% stallions; 48% Selle Français, 7% KWPN, 5% trotters and 40% other breeds; 62% showjumpers, 11% dressage, 10% eventing, and 17% other disciplines; aged from 2 to 20 years (mean ± SD, 9 ± 3 years).

### 2.2. Locomotor Examination

After collecting the anamnesis and performing the examination of the locomotor system, the veterinarian evaluated the horse locomotion without warm-up. As part of the dynamic locomotor examination, horses were trotted by their owner/groom on a straight line of 25 meters long. The handler was asked to run at adequate speed and to keep a steady pace. The ground surface was made of asphalt. Visual evaluation was performed by one of the five expert veterinarians graduated as DESV (French certification as a specialist in equine locomotor pathology) and certified ISELP (International Society of Equine Locomotor Pathology). Based on this evaluation on the straight line, horses were classified into five groups: right forelimb (RF) lame, left forelimb (LF) lame, right hindlimb (RH) lame, left hindlimb (LH) lame, and sound horses.

In total, 381 horses were screened and were evaluated lame on a straight line. Among them, 209 horses showed lameness grade ranging between 2/10 (inclusive) and 6/10 (inclusive) on a 11-grades scale equivalent to the UK scale (where 0 is: Sound and 10 is: Non-weight bearing lameness) [22,23,24]. Horses showing lameness on multiple limbs on the straight line were excluded (*n* = 33). With these criteria, 67 horses showed RF lameness, 62 horses showed LF lameness, 23 horses showed RH lameness, and 23 horses showed LH lameness. Flowchart is provided as Appendix A. Lameness grades included in each group are summarized in Table 1.

Forty-nine horses were included in the group of “sound” horses. These sound horses have been presented at the clinic for pre-purchase examination or for gait evaluation prior to further training. In this group were included individuals who met all of the following criteria (1) and (2). (1) The sound horses were in training and judged by their owners to be capable of performing all the exercises required for their sport level. (2) A full locomotor examination of these horses by an expert clinician revealed no abnormalities deemed significant under any of the examination conditions. This examination included: walk, trot on a hard circle at both reins, on a hard straight line, four flexion tests (one for each limb), trot on a soft circle at both reins. 

### 2.3. Data Collection

During the locomotor examination, as part of the clinical routine, horses were systematically equipped with the EQUISYM^®^ (Arioneo, LIM France, Nouvelle-Aquitaine, France) system consisting of seven wireless IMUs placed on the head, the withers, the pelvis, and the four cannon bones (Figure 1). They recorded tri-axial angular velocity within a range of 2000°/s and tri-axial acceleration within a range of 16 g, at a frequency of 200 Hz during approximately two trot-ups, corresponding to a mean of 14.7 ± 7.8 trot strides on a straight line. Data were recorded on the sensors and downloaded wirelessly.

### 2.4. Data Processing

First, stance phase periods, e.g., foot-on and foot-off times, were determined based on the analysis of the gyroscopic signals recorded on the four cannon bones owing to the method developed by Hatrisse et al. [25]. One stride was defined as the time between two consecutive foot-on of the left forelimb.

Then vertical displacements of the head, withers and pelvis were segmented into strides. The acceleration signal measured along the dorso-ventral axis of the horse was integrated twice and high-pass filtered using a fourth-order Butterworth filter with a cut-off frequency set to 1 Hz to obtain displacement curves [4,26]. 

Based on the vertical displacement of the head (_H), withers (_W) and pelvis (_P) occurring along a stride, four variables were calculated for each sensor location. The following asymmetry indices (AI), expressed as a percentage of the maximal range of motion within a stride, were used to compare left vs. right part of the stride (Figure 2): AI-Min was the left-right difference of the lowest point of the vertical excursion; AI-Max was the left-right difference of the highest point of the vertical excursion; AI-up was the left-right difference of the upward range of motion during the propulsion phase; and AI-down was the left-right difference of the downward range of motion during the damping phase. Positive AI value indicated a smaller movement amplitude during the right stance than during left stance, and negative AI value indicated the opposite.

All calculations were performed with custom-made Matlab2020a (The MathWorks, Natick, MA, USA) scripts.

### 2.5. Data Analysis

Mean and standard deviation (SD) were calculated from data collected in each group. Normality was assessed using graphical methods [27]. Open software RStudio (RStudio Inc., Boston, MA, USA, version 4.1.3) was used, including the packages ROCR, pROC and boot. The four AIs calculated from head, withers, and pelvis were analyzed. Receiver operating characteristic (ROC) curves were plotted to discriminate each lame limb group (RF, LF, RH, LH) from the control group (sound horses). Area under curve (AUC) of the ROC curves was calculated. Then, thresholds with highest specificity and sensitivity using the top-left method were calculated. The top-left method involves choosing the threshold related to the curve point closest to the upper-left corner of the graph. 95% confidence interval (95% CI, which values are expressed into [;] in the text) was obtained from the repartition of the best specificities and sensitivities calculated for 400 samples, using the bootstrap method based on resampling to estimate the confidence interval [28]. In this study, indices were considered having good discrimination capacity if the sum of sensitivity and specificity was strictly higher than 150% [29].

## 3. Results

### 3.1. Descriptive Results

Mean ± SD for each AI and for each horse group are summarized in Table 2 and boxplots are plotted in Figure 3. 

Means of the AIs in sound horses were close to 0% of asymmetry, particularly for the withers (AI-min = −3% ± 8%; AI-max = 2% ± 8%; AI-up = −1% ± 9% and AI-down = −3% ± 11%). The head (AI-min = −7% ± 29% and AI-up = −11% ± 29%) and pelvis (AI-min = 6% ± 8%, AI-up = 5% ± 13 % and AI-down = 5% ± 11%) showed higher absolute mean values than the withers. The negative sign of AIs for the withers and the positive sign of AIs for the pelvis expressed reduced range of movement during the LF and RH stance phase. 

RF lame horses showed higher mean values (sign of a reduced movement on the right) than sound horses for all AIs of the head and withers, and discrete lower mean values (sign of a reduced movement on the left) for all AIs of the pelvis, except AI-down_P. Like a mirror, LF lame horses showed lower mean values than sound horses for all AIs of the head and withers, and higher mean values for all AIs of the pelvis, except AI-max_H and AI-down_P.

Horses with RH lameness showed higher mean values (sign of a reduced movement on the right) than sound horses for all AIs of the head and pelvis, except AI-down_P, and showed discrete lower mean values (sign of a reduced movement on the left) for all AIs of the withers. Like a mirror, LH lame horses showed lower mean values than sound horses for all AIs of the head and pelvis, except AI-max_H and AI-down_P, and they showed higher mean values for all AIs of the withers.

### 3.2. Forelimb Lameness Discrimination

ROC curves are presented in Figure 4 for forelimbs lameness discrimination. Calculated from these ROC curves, AUC, best sensitivity and specificity, and threshold associated are summarized in Table 3. ROC curves for RF lameness discrimination showed highest AUC values of 91% [95% CI, 85;96] (AI-up_W), 90% [84;95] (AI-up_H), 90% [83;95] (AI-min_W), and 86% [78;93] (AI-min_H). The lowest AUC values were of 55% (AI-down_P), 70% (AI-min_P), 72% (AI-max_P), and 72% (AI-down_W).

As well, ROC curves for LF lameness discrimination showed highest AUC values of 91% [85;95] (AI-up_W), 79% [71;87] (AI-min_W), 79% [71;87] (AI-min_H), and 79% [70;87] (AI-up_H). The lowest AUC values were of 48% (AI-max_H), 55% (AI-min_P), 58% (AI-down_W), and 61% (AI-down_P).

For RF and LF lameness respectively, thresholds of +7% [+1;+10] and −10% [−13;−8] of asymmetry for AI-up_W resulted in 85% [77;93] and 84% [74;91] sensitivity and 88% [78;95] and 92% [84;100] specificity respectively.

### 3.3. Hindlimb Lameness Discrimination

ROC curves are presented in Figure 5 for hindlimbs lameness discrimination. Calculated from these ROC curves, AUC, best sensitivity and specificity, and threshold associated are summarized in Table 4. ROC curves for RH lameness discrimination showed highest AUC values of 88% [77;95] (AI-max_P), 86% [74;95] (AI-up_P), 84% [75;93] (AI-min_H), 81% [71;91] (AI-up_H), and 80% [69;89] (AI-down_H). The lowest AUC values were of 51% (AI-max_H), 54% (AI-down_W), 59% (AI-max_W), 62% (AI-up_W and AI-min_W).

As well, ROC curves for LH lameness discrimination showed highest AUC values of 96% [90;99] (AI-up_P), 89% [80;96] (AI-max_P), 89% [77;98] (AI-min_P). The lowest AUC values were of 48% (AI-max_H), 55% (AI-down_W), 57% (AI-down_P), and 58% (AI-down_H).

For RH and LH lameness respectively, a threshold of +18% [+16;+27] and −7% [−12;+2] of asymmetry for AI-up_P resulted in 83% [65;91] and 91% [82;100] sensitivity, and 90% [82;100] and 86% [69;91] specificity. Thresholds of +6% [+2;+8] and −10% [−19;−6] of asymmetry for AI-max_P resulted in 87% [75;100] and 78% [59;86] sensitivity, and 82% [64;88] and 88% [78;100] specificity respectively.

### 3.4. Asymmetry Thresholds of Reliable Indices

With a sum of sensitivity and specificity over 150%, head (AI-min_H and AI-up_H) and withers (AI-min_W and AI-up_W) indices discriminated the LF lame horses from sound horses. As well, head (AI-min_H, AI-max_H, AI-up_H, and AI-down_H) and withers (AI-min_W and AI-up_W) indices discriminated the RF lame horses from sound horses.

With a sum of sensitivity and specificity over 150%, withers (AI-up_W) and pelvis (AI-min_P, AI-max_P, and AI-up_P) indices discriminated the LH lame horses from sound horses. As well, head (AI-min_H and AI-up_H) and pelvis (AI-max_P and AI-up_P) indices discriminated the RH lame horses from sound horses.

Thresholds and their 95% CI associated with a sum of sensitivity and specificity over 150% for both right and left lameness discrimination were plotted in Figure 6.

## 4. Discussion

Description of AIs provided by the head, withers, and pelvis from sound horses showed that almost no asymmetry was detected on the withers, whereas head and pelvis showed slightly lower range of movement during the RF-LH stance phase. In lame horses, the withers and pelvis showed reduced range of movement during the stance phase of respectively the front and hind lame limb. The head also showed reduced movement during the stance phase of a lame forelimb. Conversely, the head showed increased movement during the stance phase of a lame hindlimb. 

Our study confirms the hypothesis that head and withers vertical displacements are indicators of forelimb lameness. Indeed, head indices (AI-min_H, AI-up_H) and withers indices (AI-min_W and AI-up_W) are the indices with the highest sensitivity and specificity (sum of sensitivity and specificity greater than 150%) for discriminating horses with forelimb lameness from sound horses. Among them, AI-up_W discriminated forelimb lameness with the highest sensitivity (>84%) and specificity (>88%). 

For hindlimb lameness, pelvic vertical displacement was the most consistent indicator. Withers and head vertical displacement were also modified, with compensatory movements but only indices from the pelvis (AI-max_P and AI-up_P) discriminated both sides hindlimb lameness with sensitivity over 78% and specificity over 82%. 

Among the four indices (AI-up, AI-max, AI-min, AI-down) used in this study, it was shown that AI-down indice has systematically a low sensitivity and specificity for discriminating both hindlimb and forelimb lame horses from sound horses. This result suggests that the AI-down indice should not be considered as the most useful indice in future work.

Main results give the following guidelines: associated with the highest sensitivity and specificity, AI-up_W discriminates LF lame horses under −10% of asymmetry and RF lame horses over +7% of asymmetry from sound horses. Associated with the highest sensitivity and specificity, AI-up_P discriminates LH lame horses under −7% of asymmetry and RH lame horses over +18% of asymmetry from sound horses. These observations confirm that the upward movement of the pelvis has the highest power to discriminate hindlimb lameness [30].

Higher relevance of the withers data than the head data contradicts previous studies [5,21]. Head shows greater movement asymmetry than withers, helping the visual assessment for forelimb lameness. However, the head is subjected to random movements existing in restless horses [7,8,10,31]. This study demonstrates that the withers movement provides useful and relevant information to detect forelimb lameness. Although having a lower reliability, head movement indices provide additional information useful to differentiate forelimb and hindlimb lameness [11,32].

Moreover, absolute threshold values of the head asymmetry were different in our study between the right and the left side of lameness discrimination (AI-up_H: +24% for RF vs. −36% for LF; AI-min_H: +6% for RF vs. −33% for LF). This difference may reflect different types of lameness between the RF and the LF lame horses in our reference population. This difference could also reflect an artefactual reduced movement of the head during the LF stance phase, compared to the RF stance phase for sound horse. Explanation could be found because horses were trotted in-hand on their left side by their owner or groom. This artefactual asymmetry may be induced by the handler despite instructions not to hold the head too firmly and to release the lunge. The head was either pulled forward, either hold backward depending on the spontaneous speed of the horse. This difference was not highlighted in other studies, which were maybe performed under more standardized conditions [33]. To a lesser extent, a similar phenomenon to the other side appeared on the pelvis in sound horses (AI-min_P mean of +6%; AI-up_P +5%; AI-down_P +5%).

Here AIs were divided by the range of movement, generating relative indices expressed in %. Values in millimeters are also provided in Appendix A. Normalizing values seems natural in order to compare movement measured in a heterogeneous population, possibly including ponies. In addition, normalizing may lead to an easier comparison of asymmetry indices processed by different gait analysis systems [15,34]. Previous studies [13,21] have however expressed thresholds in millimeters. Pfau et al. [21] found that the threshold of HDmin was 14.5 mm for forelimb lameness discrimination. As well, PDmax discriminated hindlimb lameness from 10 mm with a low sensitivity. Contrary to our results, PDmin was more reliable than PDmax, and the head was more reliable than the withers. Pfau et al. focused on specificity in a selection context (racing Thoroughbreds for the purpose of “lameness screening”) where false positives should be avoided. Conversely, in a clinical context, a fair balance between sensitivity and specificity must be found to limit both false positives (inducing unnecessary costly investigations and anxiety of the owner) and false negatives [21]. This choice may explain differences with the results of Pfau et al. [21]. Other differences were: the lack of differentiation between right and left lameness, the study of a homogeneous population, and the subjective evaluation made by five assessors using video.

In the present study, we noticed that forelimb lameness also decreased the pelvic vertical range of motion during the lame limb stance phase. This observation has been previously noticed in other studies [35,36]. LH lameness showed a small impact on the head but decreased the withers movement during RF stance phase. Contrary to LH lameness, RH lameness increased the head movement during LF stance phase. This supports the hypothesis that the head, the withers, and the pelvis provide complementary information about the lameness location [11,37].

The low number of hindlimb lame horses (23 RH and 23 LH) may induce a bias. Furthermore, all lameness were included regardless of the type of injury diagnosed. It is obvious that some indices may be more or less modified according to the type of injury. To go further, more horses and specific groups for each type of diagnosed injury or clinical manifestation will be needed. This must be in the future roadmap. 

Another limitation of this study is the clinical reference used to detect lame horses from non-lame horses and single limb lameness vs. multi-limbs lameness because this clinical assessment is recognized by definition as being subjective [2]. In the present study, visual examination of lameness by expert clinicians in the real context of the clinical examination, in the field, has been chosen as a reference to establish thresholds and calculate their sensitivity and specificity. This choice must, of course, be discussed as it is well known that visual assessment by experts is subject to many uncertainties (e.g., lack of repeatability and reproducibility) [1]. Visual assessment is not considered a “gold standard” in the present study, but only as a reference to what exists in the best possible conditions. In order for clinicians to appropriate the tools for quantifying locomotor asymmetries, it seems indeed necessary to give them an idea of the threshold values which, on these devices, correspond to what they are used to seeing and concluding subjectively with the classical (even imperfect) methods. This first step seems necessary because it is only once these benchmarks have been established that real progress can be made in interpreting the data from the quantification systems. The challenge here is to avoid the slightest asymmetry measured by a quantification system from being mistaken for lameness. It should indeed be remembered that the definition of lameness refers to a veterinarian’s diagnosis and not to a machine. The machine can only be considered as a quantitative aid to a multi-factorial medical decision. In this study, the real condition of clinical routine was deliberately chosen in order to reflect the real-life examination of lame horses. Five highly experienced veterinarians were involved. Their experience and their identical and consistent method of assessment are likely to increase the agreement rate [35,38], although this result can be discussed [1]. The agreement could for example be slightly increased if the experts had not been informed of the owner’s request. In this context, the lowest grade of lameness (1/10) was deliberately excluded because of weaker agreement for very subtle asymmetries [1].

## 5. Conclusions

Although quite small 95% CIs were found, an increased number of horses improved threshold accuracy. This study highlights the most relevant indices (AI-up_W for forelimb lameness and AI-up_P for hindlimb lameness) and indicates an order of magnitude of the thresholds and their 95% CIs. These thresholds can be used as a first support to discriminate between lame (from grade 2/10) and non-lame horses, bearing in mind the value of the 95% CIs which prohibits the use of these thresholds as an absolute cut-off value. In any case, these indicators can only be interpreted in the light of a global clinical expertise taking into account that there is not only one type of lameness but multiple clinical manifestations of locomotor disorders depending on the type of lesion. Subtle lameness (1/10 grade) have not been included here; further studies are warranted to refine the thresholds for horses with subtle lameness.

Moreover, forelimb and hindlimb lameness were analyzed separately and multi-limb lame horses were excluded. It should therefore be kept in mind that the interrelationship between the movements of the head, withers, and pelvis still requires further work. Future studies with multivariate analysis are needed to provide more information on the lame limb identification and relationship between the indices in various clinical circumstances.

In the longer term, the application of this study is aimed at a wider range of conditions in veterinary practice. A main limitation is that all measures were recorded under specific and standardized conditions. In the following years, further data are needed to refine lameness detection thresholds under conditions where physiological asymmetries are known to be higher (circles for instance) [38].

## Figures and Tables

**Figure 1 animals-12-03498-f001:**
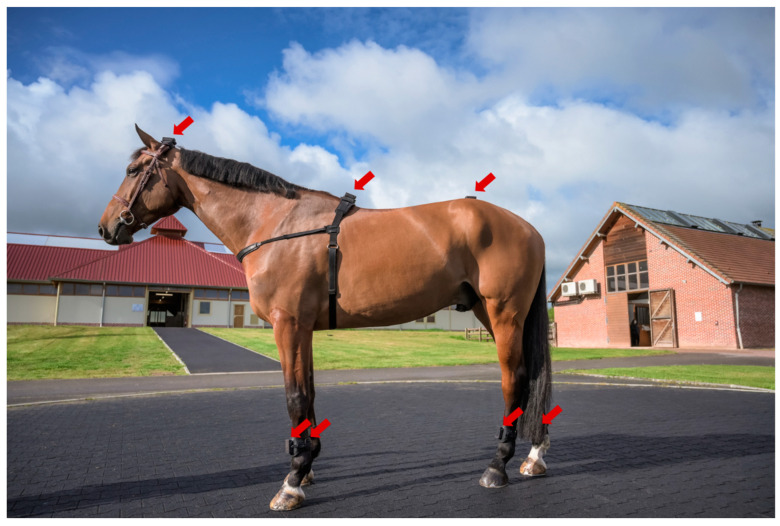
Horse equipped with EQUISYM^®^ system, composed by sensors placed on the head, the withers, the pelvis and the four cannon bones (shown by the red arrows).

**Figure 2 animals-12-03498-f002:**
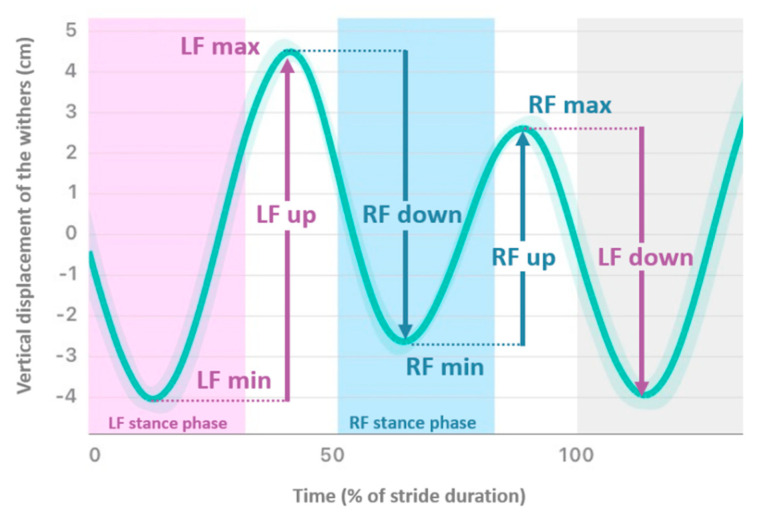
Mean vertical displacement (cm) of the withers plotted against time (expressed as % of stride duration) of a horse showing right forelimb (RF) lameness. Asymmetry Indices (AI) are: AI-min = RFmin-LFmin/LFup; AI-max = LFmax-RFmax/LFup; AI-up = LFup-RFup/LFup; AI-down = LFdown-RFdown/LFdown. (LF—Left Forelimb).

**Figure 3 animals-12-03498-f003:**
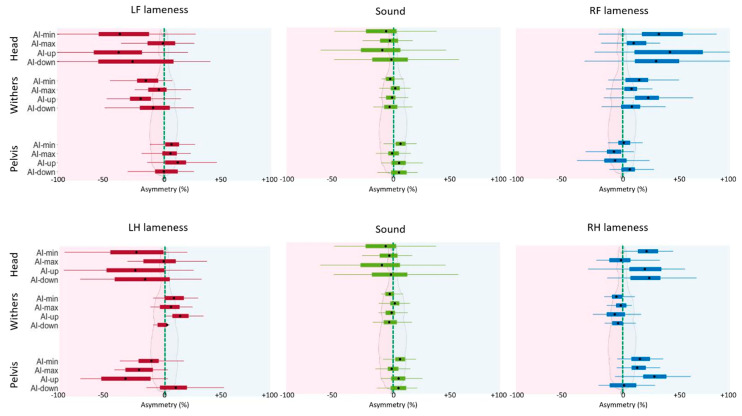
Boxplots of asymmetry indice (AI) in % for sound, left forelimb (LF), right forelimb (RF), left hindlimb (LH), and right hindlimb (RH) lame horses. The black dot represents the mean, the 2 extremities of the box are 25th and 75th percentiles, and extremities of the whiskers represent extreme data points not considered outliers. Negative value represents smaller movement on the left limb.

**Figure 4 animals-12-03498-f004:**
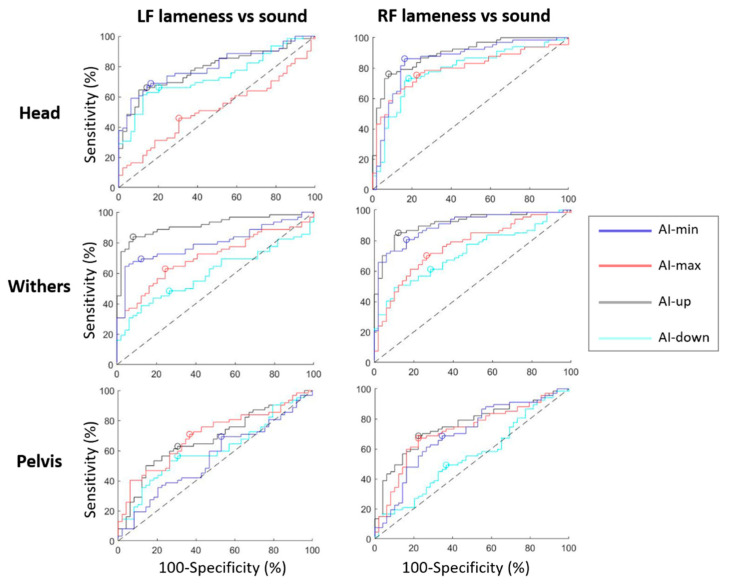
ROC curves discriminating horses with left forelimb (LF) lameness from sound horses; and discriminating horses with right forelimb (RF) lameness from sound horses, plotted for each sensor location (head, withers, pelvis); and plotted for asymmetry indice: AI-min (blue), AI-max (red), AI-up (black) and AI-down (cyan). The best specificity and sensitivity point of each curve is represented by a circle. The dashed black line is the hypothesized ROC curve with discrimination capacity only due to perfect chance.

**Figure 5 animals-12-03498-f005:**
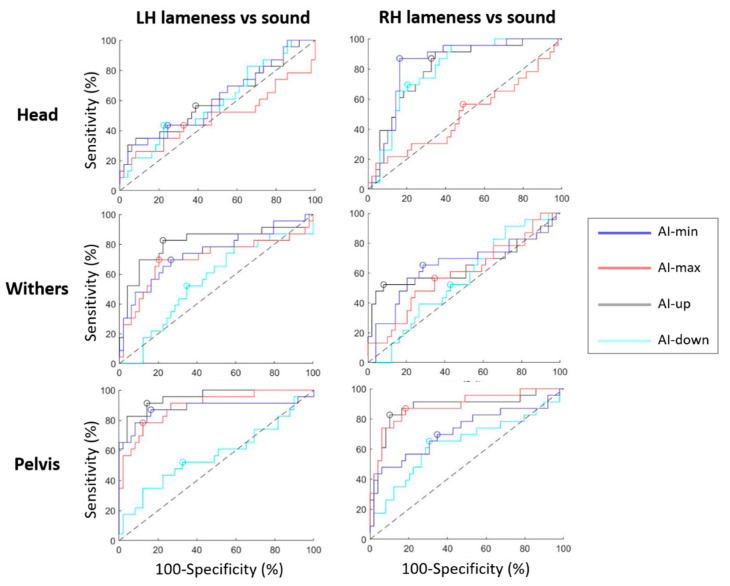
ROC curves discriminating horses with left hindlimb (LH) lameness from sound horses; and discriminating horses with right hindlimb (RH) lameness from sound horses, plotted for each sensor location (head, withers, pelvis); and plotted for asymmetry indice: AI-min (blue), AI-max (red), AI-up (black), and AI-down (cyan). The best specificity and sensitivity point of each curve is represented by a circle. The dashed black line is the hypothesized ROC curve with discrimination capacity only due to perfect chance.

**Figure 6 animals-12-03498-f006:**
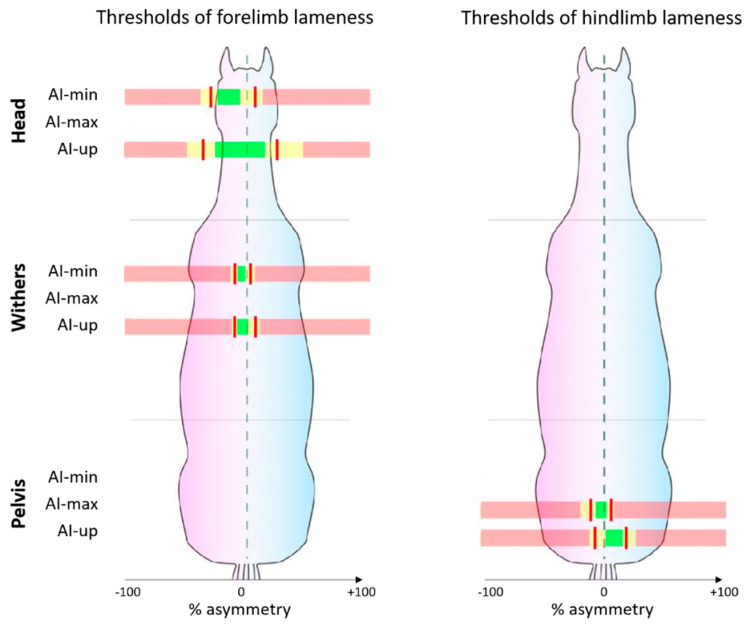
Thresholds (red line) of asymmetry indices (AI) (in % of asymmetry) for forelimb and hindlimb lameness discrimination. Only the AIs with the sum of sensitivity and specificity over 150% for both right and left lameness are plotted. Three range of values are represented: yellow for 95% confidence interval (95% CI) around the threshold, green for values below the 95% CI (“sound” horses) and red for values beyond the 95% CI (“lame” horses).

**Table 1 animals-12-03498-t001:** Number of horses showing a lameness depending on the location and the grade according to the 11-grades UK lameness scale. Mean ± SD^1^ of lameness grade in each lame horses group.

Clinical Lameness Grade	2/10	3/10	4/10	5/10	6/10	Total	Mean ± SD
Right Forelimb lameness	35	9	17	4	2	67	1.5 ± 0.7
Left Forelimb lameness	40	9	7	4	2	62	1.4 ± 0.8
Right Hindlimb lameness	7	4	11	0	1	23	1.7 ± 0.5
Left Hindlimb lameness	7	6	5	1	4	23	1.8 ± 0.6

SD^1^—standard deviation.

**Table 2 animals-12-03498-t002:** Mean ± SD of AIs of the head, the withers and the pelvis in sound, RF lame, LF lame, RH lame, and LH lame horses trotting on a hard straight line.

Location	AI^1^	Sound	RF^2^	LF^3^	RH^4^	LH^5^
Head (_H)	AI-min_H (%)	−7 ± 29	34 ± 30	−41 ± 38	22 ± 18	−27 ± 33
	AI-max_H (%)	−4 ± 12	10 ± 17	−1 ± 27	−2 ± 16	−1 ± 27
	AI-up_H (%)	−11 ± 29	44 ± 36	−42 ± 36	21 ± 21	−28 ± 36
	AI-down_H (%)	−2 ± 29	31 ± 35	−29 ± 37	25 ± 21	−19 ± 32
Withers (_W)	AI-min_W (%)	−3 ± 8	16 ± 15	−17 ± 18	−6 ± 8	9 ± 13
	AI-max_W (%)	2 ± 8	8 ± 9	−5 ± 14	−2 ± 7	6 ± 12
	AI-up_W (%)	−1 ± 9	24 ± 20	−22 ± 14	−7 ± 13	15 ± 18
	AI-down_W (%)	−3 ± 11	9 ± 14	−10 ± 24	−4 ± 7	2 ± 15
Pelvis (_P)	AI-min_P (%)	6 ± 8	1 ± 8	7 ± 10	16 ± 12	−13 ± 19
	AI-max_P (%)	−1 ± 9	−8 ± 11	6 ± 12	14 ± 10	−25 ± 16
	AI-up_P (%)	5 ± 13	−7 ± 15	13 ± 16	30 ± 17	−38 ± 27
	AI-down_P (%)	5 ± 11	7 ± 11	0 ± 14	1 ± 15	10 ± 18

AI^1^—asymmetry indice, RF^2^—right forelimb, LF^3^—left forelimb, RH^4^—right hindlimb, LH^5^—left hindlimb.

**Table 3 animals-12-03498-t003:** Area under the curve (AUC) of the receiver operating characteristic (ROC) curve discriminating sound and forelimb lame horses, for each asymmetry indice (AI). Best sensitivity, specificity, and associated threshold were calculated using top-left method of ROC analysis. [95% confidence interval] were calculated plotting ROC analysis on 400 population resamplings (bootstraps). Results for which the sum of sensitivity and specificity is over 150% for both sides are in bold.

AI	AUC RF^1^ (%)	Sensitivity RF (%)	Specificity RF (%)	Threshold RF (%)	AUC LF^2^ (%)	Sensitivity LF (%)	Specificity LF (%)	Threshold LF (%)
**AI-min_H**	**86 [78;93]**	**86 [78;95]**	**84 [73;92]**	**6 [−6;13]**	**79 [71;87]**	**69 [54;78]**	**84 [73;96]**	**−30 [−38;−24]**
AI-max_H	79 [70;87]	75 [64;86]	78 [63;87]	5 [−1;10]	48 [38;59]	46 [29;53]	69 [53;87]	−10 [−24;−4]
**AI-up_H**	**90 [84;95]**	**76 [62;81]**	**92 [85;100]**	**24 [14;45]**	**79 [70;87]**	**66 [51;76]**	**86 [74;100]**	**−36 [−49;−26]**
AI-down_H	79 [70;87]	73 [61;81]	82 [70;94]	17 [10;25]	73 [65;83]	66 [56;79]	80 [66;85]	−25 [−29;−18]
**AI-min_W**	**90 [83;95]**	**81 [68;90]**	**84 [71;93]**	**3 [−2;6]**	**79 [71;87]**	**69 [57;79]**	**88 [77;98]**	**−10 [−14;−8]**
AI-max_W	77 [68;86]	70 [56;81]	73 [59;85]	6 [3;9]	71 [61;79]	63 [48;72]	76 [65;89]	−2 [−6;0]
**AI-up_W**	**91 [85;96]**	**85 [77;93]**	**88 [78;95]**	**7 [1;10]**	**91 [85;95]**	**84 [74;91]**	**92 [84;100]**	**−10 [−13;−8]**
AI-down_W	72 [61;80]	61 [41;75]	71 [49;89]	1 [−7;7]	58 [48;69]	48 [22;58]	73 [56;100]	−8 [−16;−4]
AI-min_P	70 [59;78]	69 [56;85]	65 [45;74]	4 [1;7]	55 [43;64]	69 [59;95]	47 [18;54]	5 [−2;7]
AI-max_P	72 [63;81]	67 [55;77]	78 [66;89]	−5 [−7;−4]	70 [60;79]	71 [58;86]	63 [44;73]	1 [−2;4]
AI-up_P	76 [67;84]	69 [55;78]	78 [66;90]	−3 [−7;0]	67 [57;77]	63 [52;77]	69 [48;76]	10 [6;13]
AI-down_P	55 [44;67]	49 [32;59]	63 [48;78]	7 [4;12]	61 [50;71]	56 [47;68]	69 [53;81]	1 [−3;6]

RF^1^—right forelimb, LF^2^—left forelimb. Results for which the sum of sensitivity and specificity is over 150% for both sides are in bold.

**Table 4 animals-12-03498-t004:** Area under the curve (AUC) of the receiver operating characteristic (ROC) curve discriminating sound and hindlimb lame horses, for each asymmetry indice (AI). Best sensitivity, specificity and associated threshold were calculated using top-left method of ROC analysis. [95% Confidence Interval] were calculated plotting ROC analysis on 400 population resamplings (bootstraps). Results for which the sum of sensitivity and specificity is over 150% for both sides are in bold.

AI	AUC RH^1^ (%)	Sensitivity RH (%)	Specificity RH (%)	Threshold RH (%)	AUC LH^2^ (%)	Sensitivity LH (%)	Specificity LH (%)	Threshold LH (%)
AI-min_H	84 [75;93]	87 [73;100]	84 [73;95]	8 [4;11]	60 [45;76]	43 [6;47]	76 [64;100]	−25 [−53;−16]
AI-max_H	51 [36;66]	57 [34;70]	51 [23;63]	−5 [−14;3]	48 [32;66]	43 [17;45]	67 [47;84]	3 [−2;24]
AI-up_H	81 [71;91]	87 [74;100]	67 [47;75]	−1 [−18;7]	60 [46;75]	57 [37;73]	61 [33;78]	−19 [−42;6]
AI-down_H	80 [69;89]	70 [41;77]	80 [67;100]	16 [8;35]	58 [43;73]	43 [8;49]	78 [66;100]	−22 [−53;−14]
AI-min_W	62 [47;76]	65 [46;82]	71 [55;83]	−6 [−10;−3]	75 [61;86]	70 [50;85]	73 [57;86]	0 [−4;3]
AI-max_W	59 [43;72]	57 [37;70]	65 [46;79]	0 [−4;3]	70 [55;84]	70 [51;88]	80 [66;92]	7 [4;9]
AI-up_W	62 [46;76]	52 [29;64]	92 [74;100]	−11 [−21;−5]	81 [67;92]	83 [70;100]	78 [59;87]	3 [−6;7]
AI-down_W	54 [41;68]	52 [20;64]	57 [37;75]	−5 [−12;−1]	55 [41;69]	52 [19;59]	65 [53;90]	−7 [−13;−4]
AI-min_P	72 [58;84]	70 [53;91]	65 [33;78]	10 [1;15]	89 [77;98]	87 [76;100]	84 [68;90]	−1 [−4;5]
**AI-max_P**	**88 [77;95]**	**87 [75;100]**	**82 [64;88]**	**6 [2;8]**	**89 [80;96]**	**78 [59;86]**	**88 [78;100]**	**−10 [−19;−6]**
**AI-up_P**	**86 [74;95]**	**83 [65;91]**	**90 [82;100]**	**18 [16;27]**	**96 [90;99]**	**91 [82;100]**	**86 [69;91]**	**−7 [−12;2]**
AI-down_P	64 [49;77]	65 [47;84]	69 [54;81]	0 [−3;4]	57 [42;71]	52 [33;63]	67 [47;82]	9 [2;17]

RH^1^—right hindlimb, LH^2^—left hindlimb. Results for which the sum of sensitivity and specificity is over 150% for both sides are in bold.

## Data Availability

The data that support the findings of this study are available from the corresponding author upon reasonable request.

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
