# Peer review of "Investigation of Thresholds for Asymmetry Indices to Represent the Visual Assessment of Single Limb Lameness by Expert Veterinarians on Horses Trotting in a Straight Line"

_animals, 2022, doi:10.3390/ani12243498_

Round 1

Reviewer 1 Report

Title - should read “…in horses trotting on a straight line.”

Line 54 - I do not think biomarker is appropriate term

LIne 59 - ….IMUs), is used clinically.
LIne 64 - “complementarity” not sure this is the correct word
LIne 69 - pelvic
Table 1: the AAEP grading scale does not include half scores.  I would suggest that you describe this as a modified AAEP grading scale and describe the modifications.
Line 116 - no lameness WAS noticed
LIne 117 - these horses WERE presented at the clinic…

Discussion - what do you mean by “weaker movement”?
Line 278 - Again I don’t think biomarker is the correct term
Lines 278-280 - this sentence does not make sense, incorrect translation? I think “discriminated” should be “identified” Also I don’t think you have to say “both sides”
Line 286 - should have colon (:) after “as follows:” and then list OR rewrite so as not presenting a list
Line 323 - in other studies

Additional
Why is the total of sensitivity and specificity over 150% important?
Is + asymmetry always associated with right side and - asymmetry always left side?
You need closer inspection of your translation to english I am frequently confused by what you are trying to say.

Reviewer 2 Report

Review of manuscript “Sensitivity and specificity of asymmetry indexes to detect lameness in horses trotting on straight line”

General: Very nice to see more and more manuscripts concerning the quantitative measurement of lameness in horses and in particular the ‘expansion’ of measurements to anatomical landmarks other than the head and the pelvis (here: the withers).

The manuscript presents measurements of a number of movement symmetry parameters of head, withers and pelvis in relation to the visual assessment of the horses as ‘sound’ or as lame in different limbs and with different lameness grades (assessed visually). The emphasis of the manuscript (title + throughout the manuscript) is on ‘sensitivity and specificity’.

This is where – in my opinion – the manuscript could be further improved: the ‘gold standard’ against which sensitivity and specificity are calculated is the ‘visual lameness examination’ by experts. This visual assessment has been shown to be particularly variable (between multiple assessors, even between experts, as cited by the authors [1]) and the ability of humans to reliably detect movement asymmetry also appears to be limited. As such all studies aiming at establishing ‘thresholds’ for determining at which point an ‘asymmetry becomes a lameness’ are limited and calculating sensitivity and specificity values will suffer from in particular the variability at the lower end of the lameness scale.

The authors have generally done a good job of navigating this issue by discussing it. Here are some items that could do with a little more attention:

·       -in particular horses with hind limb lameness have been found to show comparably high compensatory head movement asymmetries [2] and these compensatory changes are also measurable after diagnostic anaesthesia [3]. In that respect it would be useful to:

o   learn more about how it was determined that the horses were indeed unilaterally lame (no multi-limb lameness present) and how the conclusion about soundness was reached (which exercise conditions etc were used for this).

o   how many horses were ‘screened’ for inclusion into the study and then excluded due to multi-limb lameness. In the reviewers own experience (in referral level lameness), in many horses with mild lameness more than one limb may be involved in the lameness. (and also address the exclusion of grade 0.5 lame horses)

·       When looking at table 2 and figure 3 it is striking that the SD values for the ‘sound’ horses are rather high? This may be related to above mentioned ‘issues’ (inclusion/exclusion criteria) and these rather large SD values are very likely related to the comparatively low sensitivity and specificity values in particular for head movement asymmetry values? Again, this demands a more detailed description of how the ‘sound’ horses were evaluated and needs a more balanced discussion about the ability of humans to detect movement asymmetry and agree about lameness grades in particular at the lower end of the lameness scale, which is the end of the scale that is of particular importance here when aiming to investigate sensitivity and specificity for differentiating between ‘sound’ and ‘lame’.

·       It is interesting to note that in the discussion the authors argue that ‘withers data’ has ‘higher relevance’ than ‘head data’ for forelimb lame horses and they mention that ‘the head is also impacted by hindlimb lameness and (is) subjected to random movements’:

o   This sentence needs some references for example to studies quantifying compensatory movements and to studies describing methods of dealing with ‘random movements’ (for example [4] for the latter). Multiple studies describe compensatory effects.

o   This sentence also manages to almost derail the whole study with the authors arguing that hind limb lameness may be relevant for explaining the lower sensitivity and specificity of head movement asymmetry for forelimb lame horse in the presented study (so are the horses truly unilaterally lame?): from figure 3 it becomes evident that there may indeed be horses in this cohort that are here classified as ‘forelimb lame’ but the head movement of some of these horses indicates a head movement asymmetry that is near symmetrical or opposite to what would be expected in these horses, were they truly unilaterally forelimb lame (i.e. RF type asymmetry in horses classified LF lame or vice versa). For example, it is impossible to tell without having access to individual horse data in how many horses this a ‘contra-lateral’ head/withers asymmetry may have been present. The authors are encouraged to provide information to further show how many of the ‘forelimb lame’ horses adhere to the expected pattern (same asymmetry head and withers) and also how many of the hind limb lame horses show the expected ‘contra-lateral’ head/withers pattern.

·       It is also interesting to note that the authors indicate that the AAEP lameness scale has been applied and at the same time use ‘half grades’ on a lameness scale that has ‘distinct criteria’ assigned to each ‘full’ grade. What is a 2.5 lameness then? How many of the features of grade 2 and how many of grade 3 have to be fulfilled? Have all horses been assessed under different conditions in addition to the straight line trotup that has been measured so as to being able to assess how consistent the lameness is (one of the criteria of the AAEP scale). Or is it rather the case that a numerical scale form 0 to 5 has been used (much like to UK 0 to 10 scale just using half grades)?
Very importantly, right at the end of the manuscript, the authors indicate that horses with a lameness grade of 0.5 on the 0 to 5 scale have been excluded. This needs to be presented much more prominently. Naturally, when studying sensitivity and specificity, excluding subjects close to the decision border (here: sound versus lame) will more or less heavily influence the overlap between the two groups and as such can have a rather large influence on the sensitivity and specificity values as well as the ‘threshold’ values chosen from the ROC curves. This exclusion of horses near the decision boundary with a 0.5 lameness grade needs to feature more prominently (abstract, materials and methods, discussion). As it is presented currently, it is presented as one of the last sentences of the manuscript.

·       While I fully understand the desire to ‘normalize’ the movement asymmetry values and here use percentage asymmetry values, this may have rather complex consequences. In the context of being able to compare the results of the present study to previous studies on movement asymmetries using ‘non-normalized’ values it seems (as a minimum requirement) necessary to provide additional information, such as the normalization values used for each horse (i.e. the range of motion). For example add a supplementary data table with information for each horse including the asymmetry values and the normalization values used for each horse.
It would be interesting to evaluate the consequences of using non-normalized values in particular with respect to the amount of variation of head movement parameters in the sound horses: how can horses with >50% or <-50% asymmetry in head movement be classified as ‘sound’? Do these horses have a particularly small range of motion? Or do they have large asymmetry components that the observers deemed irrelevant in the context of lameness grading?

Abstract:

·       Please provide threshold values in addition to sensitivity and specificity values.

·       Please consider adding the use of ‘visual lameness scoring’ as the ‘gold standard’ for calculating sensitivity and specificity to the main limitations of this study.

·       Mention the exclusion of horses with lameness grade 0.5 (out of 5) as a major limitation for the sensitivity and specificifity and threshold values presented here.

Introduction

·       “The issue about the relationship between visually qualitative and measured quantitative lameness assessment has been then raised”. I would argue that these systems are not ‘measuring lameness’, but they are measuring gait features that are typical signs of lameness such as asymmetry of upper body movement in trotting horses on the straight.

·       “They were based on a confidence interval calculated with two repeated measures on 236 horses [14].” It is my understanding that the thresholds are derived with some sort of regression analysis and that it is ‘merely a coincidence’ that the repeatability values in [14] correspond to these. I am not aware that details about the actual method used for calculating the thresholds have been published? If so, please consider adding more details.

·       “Recently, a discrimination method of statistical analysis was applied on a small sample of horses to redefine higher thresholds (14.5 mm for the head and 7.5 mm for the pelvis) [21].” In my opinion, it is important to note that the referenced study had been conducted in racing Thoroughbreds for the purpose of ‘lameness screening’ in a horseracing context where specificity was emphasized. In the clinical lameness examination, in contrast, a high sensitivity is typically desirable when identifying the most likely affected limb in a lame horse. As such ‘threshold values’ for the two scenarios can be expected to be very different.

Materials and Methods

·       What is a “static locomotor examination”? Was the horses ‘static’ (not moving) or was it a locomotor examination, i.e. in movement? Maybe this is referring to the examination of the 'locomotor system' (limbs etc)?

·       “Visual evaluation was performed by one of the expert veterinarians graduated as specialist in equine locomotor pathology”: board certified ECVS/ACVS or ECVSMR/ACVSMR?

·       “Horses showing lameness on multiple limbs were excluded.” See comment about specifying about how many horses were evaluated and how many included in the study and specifc commen about 0.5 lameness grade!

·       “With these criteria, 67 horses showed RF lameness, 62 horses showed LF lameness, 23 horses showed RH lameness and 23 horses showed LH lameness. Lameness grades included in each group are summarized in Table 1.” Nice to have an almost balanced distribution of horses into groups of left and right lame horses. Was this inclusion performed ‘prospectively’ or ‘retrospectively’ (see also comment about how many horses screened and how many excluded for multi-limb lameness).

·       “No lameness has been noticed by the veterinarian through the full lameness examination in 49 horses” Can you provide more information about which conditions this ‘full examination’ included?

·       Emphasize again the exclusion of 0.5 grade lame horses. It seems curious that this exclusion has been performed since in the context of detecting the lame limb in a horse undergoing a lameness examination demands the use of a method with a high sensitivity. Specificity is of lesser importance at the initial stage since the follow-up steps of the lameness examination, lunge, flexion, possibly ridden exercise and then of course diagnostic analgesia will provide opportunity to rule out initial ‘false positives’.

·       “They recorded tri-dimensional angular velocity within a range of 2000°/s and tri-dimensional acceleration within a range of 16g, at a frequency of 200 Hz during a mean of 14.7 ± 7.8 trot strides on a straight line.” Tri-axial may be a better word? 14.7 strides seems comparatively low. How many trotups were used per horse? Are data transmitted wirelessly and in real-time to a receiver (tablet/laptop) or recorded on the sensors and downloaded wired/wirelessly?

·       “… weaker movement amplitude …”. Maybe a ‘reduced movement amplitude’. The force related to the movement is ‘weaker’, however the movement itself is ‘smaller’ or ‘reduced’.

·       “Mean and Standard Deviation (SD) were calculated from data collected in each group”. Were data assessed for normality?

·       “… using the top-left method … ” Can you provide more details about how this works?

·       “95% confidence interval (95% CI) was obtained from the repartition of the best specificities and sensitivities calculated for 400 samples created from the data collected (bootstraps)”. Nice to have confidence intervals. What are ‘samples’ in this context? This is not immediately clear here?

Results:

·       “RF lame horses showed higher mean values than sound horses for all AIs of the head and withers, and lower mean values for all AIs of the pelvis, except AI-down_P. Like a mirror, LF lame horses showed lower mean values than sound horses for all AIs of the head and withers, and higher mean values for all AIs of the pelvis, except AI-max_H and AI-down_P. Horses with RH lameness showed higher mean values than sound horses for all AIs of the head and pelvis, except AI-down_P, and showed lower mean values for all AIs of the withers. Like a mirror, LH lame horses showed lower mean values than sound horses for all AIs of the head and pelvis, except AI-max_H and AI-down_P, and they showed higher mean values for all AIs of the withers.”

Here it would be nice to add information about how many of the forelimb (and of the hind limb) lame horses showed ipsilateral/contralateral “head-withers” asymmetries (according to Rhodin et al as an additional quantitative aid for differentiating ‘true’ from ‘compensatory’ head nod and or “head/pelvis” asymmetries (i.e. in accordance with the “law of sides”: ipsilateral head/pelvis in hind limb lame, contra-lateral head/pelvis in forelimb lame. Adding a supplementary table with asymmetry values (including the range of motion values used for normalization) for each horse would help here.

·       Figure 3: very nice illustration of the amount and direction of movement asymmetries across the body of the horse!
Why are the box plots showing mean values (assuming normal distribution) and interquartile ranges (not assuming normal distribution)?

·       The remainder of the results section is somewhat hard to digest: lots of numbers and different types of brackets () [] (can this be simplified?). Can this be presented in a more intuitive manner: is it possible to add some ‘summative sentences’ for example summarizing the results according to anatomical location (head; withers; pelvis) or according to the type of asymmetry (min; max; up; down)?

·       Table 3 and 4: nice summary. Is it possible to ‘highlight’ the parameter with the best discriminatory power (highest AUC or highest sensitivity/specificity combination) somehow? Using bold type face or color etc

·       Figure 4: interesting to see that AI-min for the pelvis is not very useful for LF lameness but more useful for RF lameness. Is this a reflection of ‘different types’ of forelimb lameness between LF and RF lame horses: LF lame horses not showing AI-max component (pushoff asymmetry) while RF lame horses do?

·       Figure 5: it is very interesting to note that LH lame horses do NOT show evidence of consistent compensatory head movement asymmetry while RH lame horses do (for anything but AI-max head). Also interesting to note that the AI-down parameters (with the exception of head movement in RH lame horses) do not appear to be very useful?

·       Similar to comments for figure 4: are these different ‘types’ of hind limb lameness between LH and RH groups? AI-min pelvis (weight bearing component, [5]) is more useful for LH lame horses and much less so for RH lame horses. Does this potentially relate to the different head movement asymmetry patterns? And/or how many LH and how many RH lame horses for example show contralateral head/withers asymmetry (hence more likely to be unilaterally hind limb lame)?

·       Section 3.4, first paragraph: can this somehow by described a little more: these are all head and withers parameters but one needs to inspect all acronyms to really get this information.

·       Section 3.4, second paragraph: same here, these are different parameters for LH and RH lame horses, state explicitly which anatomical location and/or asymmetry parameter types (min max up down) are ‘useful’.

Discussion

·       “In lame horses, the withers and pelvis showed weaker movement during the stance phase of respectively the front and hind lame limb. The head also showed weaker movement during the stance phase of a lame forelimb” Suggestion to replace ‘weaker’ with ‘reduced’ or ‘smaller’?

·       “with a sensitivity added of specificity over 150%.” Rephrase: this phrase makes no sense? ‘added of’?

·       “Higher relevance of the withers data than the head data contradicts previous studies [5,21]. Head shows greater movement asymmetry than withers, helping the visual assessment for forelimb lameness. However, the head is also impacted by hindlimb lameness and subjected to random movements existing in restless horses. This study demonstrates that the withers movement provides complementary and reliable information compared to the head in order to detect forelimb lameness.”
See previous ‘general comment’. This is really where it needs to be made clearer that withers asymmetry has been found to be useful in conjunction with head movement (going back to [6,7]). Withers movement is affected by BOTH forelimb and hind limb related movement asymmetry.
Essentially the argument brought forward here that head movement is affected by hind limb lameness makes little sense here since the authors explicitly indicate that multi-limb lame horses were excluded, so the potential of not all included horses being unilaterally forelimb (or unilaterally hind limb) lame needs to be discussed here when mentioning this line of argument. And more information about the different groups of horses is needed (individual horse data in supplementary table, number of horses showing ipsi/contra-lateral head/withers asymmetries).

·       “Moreover, absolute threshold values of the head asymmetry were different in our study between the right and the left side of lameness discrimination (AI-up_H: +24% RFvs -36% LF; AI-min_H: +6% RF vs -33% LF).“ Another option being that the distribution of horses with different ‘types’ of lameness (weight bearing / pushoff, true forelimb/ compensatory) etc is different between left and right lame horses (of different grades)? There is an abstract published that presented some evidence about the ‘side of handling’ (no evidence found for side of handling significantly affecting movement asymmetry, small number of horses), but unfortunately there does not seem to be a full manuscript on this. (Effect of Side of Handling on Movement Symmetry in Horses, https://doi.org/10.1111/evj.12267_127)

·       “Here AIs were divided by the range of movement, generating relative indexes (indices?) expressed in %, whereas previous thresholds of the absolute difference of minimum “HDmin” and maximum “HDmax” of the head, and absolute difference of minimum “PDmin” and maximum “PDmax” of the pelvis, have been expressed in millimeters [13]. Pfau et al. (2020) found that the threshold of HDmin was 14.5 mm for forelimb lameness discrimination [21]. As well, PDmax discriminated hindlimb lameness from 10 mm with a low sensitivity. Contrary to our results, PDmin was more reliable than PDmax, and the head was more reliable than the withers. However, Pfau et al. (2020) did not separate right and left lameness, and the subjective evaluation was made using video. Hardeman et al. (2022) consider that the assessment of lameness using video can underestimate the grade [26].”
Several remarks here:
consider ‘unnormalizing’ your values here either based on average values for range of motion or ideally based on individual horse values for a more direct comparison (and for later meta-analysis between studies/publications).
In this context it needs to be considered that the current manuscript is concerned with clinical examination of horses and identifying thresholds for this application while the cited manuscript is about ‘screening’. In clinically lame horses, the task is to identify the affected limb which requires high sensitivity (and lower specificity is okay since it has already been determined that the horse indeed has a problem hence is presented for a lameness exam) while a screening of racehorses requires a high specificity (low number of false positives) and hence different thresholds can be expected for these different scenarios. In addition, the cited study uses racing Thoroughbreds in active race training and high intra- and inter-day variations have been found for these. Again, the exclusion of 0.5 grade lame horses here is curious.
Also, a previous study has found high inter-rater disagreement also for ‘live’ workups of lame horses, so yes video assessment may matter but also ‘live’ assessments are variable between observers [1] and the cited study in Thoroughbreds used ‘agreements’ between multiple observers.

·       “Their experience and their consistent method of evaluation raised the agreement rate”. See [1] for a more balanced argument on increased agreement with experienced observers.

·       “Moreover 0.5/5 grade lameness were excluded because of weaker agreement for subtle lameness [1].” This really is a “bomb shell” at the end of the manuscript: this absolutely needs to be stated explicitly in the materials and methods. It was not clear until this point that mildly lame horses (essentially 1/10 lame horses) were excluded from the study!

Conclusions

“This study highlights the most relevant indexes (AI-up_W for forelimb lameness and AI-up_P for hindlimb lameness) and indicates an order of magnitude of the thresholds and their 95% CIs. These thresholds can be used as a first support to discriminate against lame and non-lame horses, bearing in mind the value of the 95% CIs which prohibits the use of these thresholds as an absolute cut-off value. In any case, these indicators can only be interpreted in the light of a global clinical expertise taking into account that there is not only one type of lameness but multiple clinical manifestations of locomotor disorders depending on the type of lesion.”

In the context of the previously published manuscripts on the role of withers movement for forelimb and hind limb lameness, it may be indicated to mention the interrelation of head and withers in this context.
Also suggest to replace “discriminate against” by “differentiate between”?

references

1.           Keegan, K.G., Dent, E.V., Wilson, D.A., Janicek, J., Kramer, J., Lacarrubba, A., Walsh, D.M., Cassells, M.W., Esther, T.M., Schiltz, P., Frees, K.E., Wilhite, C.L., Clark, J.M., Pollitt, C.C., Shaw, R. and Norris, T. (2010) Repeatability of subjective evaluation of lameness in horses. Equine Veterinary Journal 42, 92–97.

2.           Rhodin, M., Pfau, T., Roepstorff, L. and Egenvall, A. (2013) Effect of lungeing on head and pelvic movement asymmetry in horses with induced lameness. Veterinary journal (London, England : 1997) 198 Suppl, e39-45.

3.           Maliye, S. and Marshall, J.F. (2016) Objective assessment of the compensatory effect of clinical hind limb lameness in horses: 37 cases (2011-2014). American Journal of Veterinary Research 249, 940–944.

4.           Keegan, K.G., Pai, P.F., Wilson, D.A. and Smith, B.K. (2001) Signal decomposition method of evaluating head movement to measure induced forelimb lameness in horses trotting on a treadmill. Equine veterinary journal 33, 446–451.

5.           Bell, R.P., Reed, S.K., Schoonover, M.J., Whitfield, C.T., Yonezawa, Y., Maki, H., Pai, P.F. and Keegan, K.G. (2016) Associations of force plate and body-mounted inertial sensor measurements for identification of hind limb lameness in horses. American Journal of Veterinary Research 77, 337–345.

6.           Persson-Sjodin, E., Serra Braganca, F., Pfau, T., Egenvall, A., Weishaupt, M. and Rhodin, M. (2016) Movement symmetry of the withers can be used to discriminate primary forelimb lameness from compensatory forelimb asymmetry in horses with induced lameness. Equine Veterinary Journal. Supplement 48, Suppl., 32.

7.           Rhodin, M., Persson-Sjodin, E., Egenvall, A., Serra Braganca, F., Pfau, T., Roepstorff, L., Weishaupt, M., Thomsen, M.H., van Weeren, P.R., Hernlund, Elin, Braganca, F.M.S., and Egenvall, A. (2018) Vertical movement symmetry of the withers in horses with induced forelimb and hindlimb lameness at trot. Equine veterinary journal 50, 818–824.

Reviewer 3 Report

The aim of the study is good “Defining whether a gait asymmetry should be considered as lameness is challenging. Gait analysis systems now provide relatively accurate objective data, but their interpretation remains complex. Thresholds discriminating lame from sound horses and locating the lame limb with precise sensitivity and specificity are essential for the asymmetry measure interpretation.”

The present study investigates the sensitivity and specificity of different asymmetry indexes for lameness detection generated from IMU sensors. To use visual assessment as the gold standard can´t  be recommended due to the known low inter-rater agreement especially for low degree lameness (Keegan et al. 2010). This is the reason why objective tools for lameness quantification have been developed.

It is already known that the objective lameness measurements are more accurate than the visual assessment of vertical movement asymmetries in trotting horses. There is also enough evidence of the capacity of theses asymmetry indexes to correctly detect fore-and hindlimb lameness (Serra Bragança et al. . On the Brink of Daily Clinical Application of Objective Gait Analysis: What Evidence Do We Have so Far from Studies Using an Induced Lameness Model?)

The difficult part is to decide when movement asymmetries are related to pain/pathology and a clinical problem and when they are not but that requires another study design. 

In the present study the results are more reflecting the clinicians ability to assess lameness. We also don´t know if the horses were classified correctly by the clinician since no diagnostic analgesia was performed to confirm if the identified lame limb was correct. The accuracy of the visual assessment will affect the calculated sensitivity and specificity values of the objective measures. Horses showing lameness on multiple limbs were excluded. From the visual assessment, it can be difficult to know if there is true multi limb lameness or if there is compensatory movement patterns in single limb lame horses resembling lameness in the other body half. We know that approximately 30% of single hindlimb lame horses show a very prominent head nod resembling a forelimb lameness.

To look at the sensitivity and specificity for the different asymmetry indexes to discriminate forelimb lame horses from sound horses are not enough since you also want to discriminate them from hindlimb lame horses where both withers and head movement asymmetries can be present.

What is still lacking is the sensitivity and specificity for when movement asymmetries are associated to pain since there is a large amount of horses presented with movement asymmetries in different populations of horses in training and we don´t know if they are all caused by pain.

I also question the lameness scale used in the present study since for AAEP grade 1 lameness would not be visible on the straight line. In the present study 89 horses showed 1 degree lameness (AAEP) on the straight line and 68 horses between 1.5 and 2 degrees, see definition on the AAEP scale below.

AAEP-scale

1: Lameness is difficult to observe and is not consistently apparent, regardless of circumstances (e.g. under saddle, circling, inclines, hard surface, etc.).

2: Lameness is difficult to observe at a walk or when trotting in a straight line but consistently apparent under certain circumstances (e.g. weight-carrying, circling, inclines, hard surface, etc.).

What is the sensitivity and specificity for the different clinicians to detect lameness in the present study? It has been shown that more experienced veterinarians tend to see hindlimb lameness in sound horses. Starke and Oosterlinck 2018 concluded “Visual gait assessment may overall be unlikely to reliably differentiate between sound and mildly lame horses irrespective of an assessor’s background”.

Line 182 Horses with RH lameness showed higher mean values than sound horses for all AIs  of the head and pelvis, except AI-down_P, and showed lower mean values for all AIs of the withers.

Did they show lower mean values of the withers compared to sound horses? Was the absolute values compared? Hindlimb lame horses do often show a withers movement asymmetry but of opposite side of the hindlimb lameness. It seems a little bit strange that the withers values are lower than the sound horses. The sound horses would have very low values on the withers symmetry measures. It looks from table 2 that the withers AI_min are higher.

Line 291 These observations confirm that the upward movement of the pelvis has the highest power to discriminate hindlimb lameness [25].

There is a problem here since the clinician do not discriminate impact hindlimb lame horses from horses with push-off lameness. An impact lameness will give a high AI_min_P  and a push-off lameness a high AI_max_P. If there is a mixture of these types of lameness in the data set the upward movement of pelvis will be the best indicator since it changes with both types of lameness. However, there is a problem to use the upward movement when horses has an impact lameness on one hind limb and a push-off lameness on the other hindlimb since then the upward movement of pelvis can not be used to detect lameness. Therefore in the clinical setting the use of AI_min_P and AI_max_P separately is to be recommended.

Depending on the type of hindlimb lame horses included in the study (impact or push-off) the AI_min_P and AI_max_P will have different sensitivity and specificity since the clinician evaluate them both as only hindlimb lame. Between clinicians there is a big variation in the skills to detect a push-off lameness.

Round 2

Reviewer 1 Report

I am happy with improvements.  My only final critique was Line 113 - please write out 33 as it is the beginning of the sentence.  I did not identify any other concerns.

Reviewer 2 Report

Thank you for addressing the majority of comments of the first round.

There are a number of comments remaining to the revised version of the manuscript

I see that the authors disagree about the point raised about normalized and non-normalized data. While I still disagree and think that investigating this would be interesting, I accept that the authors disagree. However, I think in the context of providing the means for later ‘meta studies’ being able to combine movement asymmetry data of multiple studies, it seems mandatory that the authors provide a complete list of per horse movement asymmetry data providing the assigned lameness grade, all the movement asymmetry values AND the values used for normalizing the data (i.e. the range of motion values for the different anatomical landmarks) in order that other researchers can ‘unnormalize’ the data for a meta-analysis. This table must be provided as supplementary information.

Introduction

“Recently, a discrimination method of statistical analysis was applied on 25 Thoroughbred racehorses to redefine higher thresholds, emphasizing specificity (14.5 mm for the head and 7.5 mm for the pelvis) [21]. These results have given guidelines but require confirmation by further investigations with heterogeneous horses and lameness types using a clinical environment faced by practitioners.”
Thank you for making these changes. I feel that the changes could be slightly more ‘specific’ and emphasize that ‘screening’ (for e.g. for Thoroughbred racehorses and clinical diagnosis of the cause of a lameness are completely different in the sense of what sensitivity and specificity values are required.

Materials and methods:

Thank you for addressing the previous questions about the lameness grading (AAEP scale) and the exclusion of horses with 0.5 (out of 5) lameness.

“33 horses showing lameness on multiple limbs on the straight line were excluded”.
I feel like this would an ideal opportunity to also mention that horses with grade 1 unilateral lameness were excluded. It is mentioned in the abstract and mentioned later when the inclusion/exclusion is described in more detail but seems like a missed opportunity to already mention this here.

Thanks for adding more details about how ‘soundness’ was evaluated.

Results

Thanks for the implemented changes. Please add a supplementary table with the horse by horse data including the range of motion values.

Discussion

“Normalizing values seems essential in order to compare movement measured in a heterogeneous population, possibly including ponies. In addition, normalizing may lead to an easier comparison of asymmetry indexes processed by different gait analysis systems [15,34]”
I don’t think ‘essential’ is the correct word here. Another way of thinking about this is to provide the actually measured asymmetries together with the normalization values and additional information about for example age, breed, sex, height at withers etc. Please rephrase.

“Conversely, in a clinical context, a fair balance between sensitivity and specificity must be found to limit both false positives and false negatives [21].”
One may also argue that the ‘clinical context’ needs to be further differentiated into a lameness examination in a horse with a confirmed ‘issue’ where one of the first tasks is to identify the most likely affected limb and this needs a high sensitivity and not a high specificity since further tests will be conducted after that initial assessment. Second in a clinical context, a prepurchase examination may require different tradeoffs between sensitivity and specificity.

Thank you also for expanding the discussion on the use of visual assessment as a reference for calculating sensitivity and specificity. I must admit I find the discussion of indicating that it is not a ‘gold standard’ slightly ‘ironic’ since mathematically for calculating sensitivity and specificity the lameness grades are used to differentiate between lame and sound horses so as a ‘gold standard’ in the context of the ROC curves and then this ‘gold standard’ is used in a way that is beneficial for calculating ROC curves by omitting a lameness grade and hence likely reducing the overlap between groups.

Okay, this is, where the current argument gets really confusing: “Ideally, agreement between blinded experienced veterinarians could reduce the subjectivity of this assessment. In this study, the real condition of clinical routine was deliberately chosen in order to reflect the real-life examination of lame horses. Five highly experienced veterinarians were involved. Their experience and their identical and consistent method of assessment are likely to increase the agreement rate [35,39], although this result can be discussed [1]. Moreover 1/10 grade lameness was excluded because of weaker agreement for subtle lameness [1].”
So the current way of arguing says: ‘blinded’ to what? (are we talking diagnostic analgesia now? Arkell et al ,2006)? We need to reduce the subjectivity, but actually because we use highly experienced clinicians this is not so important, but we do not trust the highly experienced clinicians and hence we exclude mildly lame horses. Is it not exactly this decision boundary between sound and mildly lame horses that needs more investigations and yes the sensitivity and specificity value would be lower but would it not be interesting to further investigate exactly this? To play devil’s advocate: if we simply show that quantitative gait analysis agrees very well with subjective evaluation, why even bother with the measurements?

Table 3: last sentence of table descriptor: I assume this should be ‘sum of sensitivity and specificity’?

Table 4: same as above.

Reviewer 3 Report

Dear Authors,

I don´t think you can calculate sensitivity and specificity with no gold standard when it is known that the sensitivity for the objective measures is higher compare to clinicians (Comparison of an inertial sensor system of lameness quantification with subjective lameness evaluation, McCracken et al.).

 Especially not when the clinician and sensors use different levels of defining lameness.  The clinicians define if the horses is hindlimb lame or not while the symmetry measures differentiate impact (PDmin) hindlimb lame horses from horses with pushoff (PDmax) lameness. Depending on the distribution of impact/pushoff lame horses in the case load this will highly influence the calculated  sensitivity for these variables. It is also not known if the clinicians included in the study have the skills to detect a pushoff hindlimb lameness?

 The inter-rater agreement for assessing lameness and defining soundness has repeatedly been shown to be poor. In the study by Keegan et al. 2010 the inter-rater agreement for lameness < 1.5 degree lameness was low (ĸ 0.32 and 0.14 for the forelimbs and hindlimbs, respectively). In the study by Hammarberg et al. the inter-rater agreement was poorest (ĸ 0.08) for defining the horses as sound or not during locomotion on a circle.

In the following study (Comparison of visual lameness scores to gait asymmetry in racing Thoroughbreds during trot in-hand. Pfau et al.) six veterinarians evaluated lameness in 25 horses. In two horses all veterinarians agreed on soundness. For the other 23 horses they were all assessed as sound and lame by the different veterinarians, varying between 0-4 degrees between veterinarians on a 0-5 lameness scale. This means that all veterinarians have very different thresholds for when they think a horse is sound. Therefore, the visual assessment cannot be used to calculate sensitivity and specificity for the different symmetry measures from the IMUs.

There are several studies already comparing visual lameness assessment in comparison to quantitative gait analysis data in horses. Hardeman et al., McCracken et al., Pfau et al.

Is the IMU-system used validated? If not this needs to be done, both hardware and signal processing algorithms, and compared to already validated IMU or optical motion capture systems. Is the cut-off frequency for the filter optimal in relation to the stride frequency of the horses included? See reference Quantitative lameness assessment in the horse based on upper bodymovement symmetry: The effect of different filtering techniques on the quantification of motion symmetry F.M. Serra Braganca

If you want to publish your data you can compare the objective data to the visual assessment and define different lameness thresholds for the different veterinarians in your study and also look at their ability to detect hindlimb lameness in horses with either impact, pushoff or combined impact-pushoff lameness.
